

# Detection of O25B-ST131 clone and $bla_{\text{CTX-M-15}}$ gene in *Escherichia coli* isolated from patients with COVID-19

Khanda Abdulateef Anwar

Department of Basic Medical Sciences, College of Medicine, University of Sulaimani, Sulaimaniyah, Iraq

Corresponding author
Khanda Abdulateef Anwar,
khanda.anwar@univsul.edu.iq

## ABSTRACT

**Background**. Isolation of $bla_{\text{CTX-M}}$ family of extended-spectrum beta-lactamases (ESBL) is a challenge in the field of microbiology in our locality that makes treatment fail and disseminate quickly.

**Objectives**. To determine the prevalence of $bla_{\text{CTX-M-15}}$ ESBL gene in *Escherichia coli* clone O25B-ST131 isolated from COVID-19 patients with different infections.

**Methods**. This cross-sectional study was conducted on 528 patients hospitalized due to COVID-19 infection with various symptoms from April to September 2021. Using standard culturing techniques, *E. coli* were isolated from patients' various samples (urine, catheter tip, sputum, blood, endotracheal tube aspiration, pleural/peritoneal fluids, and throat swab). After the antibiotic susceptibility test, *E. coli* isolates that were resistant to more than one of the three cephalosporins (cefotaxime, ceftriaxone and ceftazidime) were tested for ESBL production using the double disc synergy test and combined disc test, then confirmed by genotypic detection of $bla_{\text{CTX-M-15}}$ gene among clones of O25B-ST131 *E. coli*. Finally, it was sequenced and its incision number was received from NCBI.

**Results**. A total of 234 *E. coli* isolates were detected from various patients' samples, and all isolates showed multiple degrees of antibiotic resistance, especially ceftriaxone, ceftazidime, and cefepime. The phenotypic test showed that 63.2% of *E. coli* isolates were positive for ESBL, of which 58.1% were confirmed by double disc synergy test (DDST) ($p = 0.002$), 83.8% by combined disc test (CDT1) ($p < 0.001$) and 60.1% by CDT2 ($p < 0.001$). However, CDT1 has a better agreement as a phenotypic screening test (72.5% with a kappa value of 0.24) than DDST and CDT2. Most *E. coli* isolates were positive for the $bla_{\text{CTX-M-15}}$ gene (68.4%), of which 75% were positive for the O25B-ST131 clone.

**Conclusions**. Most *E. coli* isolates were ESBL producers, held $bla_{\text{CTX-M-15}}$ gene and were positive for the O25B-ST131 clone.

## INTRODUCTION

*Escherichia coli* (*E. coli*) is a Gram-negative bacillus that is a part of normal intestinal flora, but it can cause intestinal and extra-intestinal disease in humans. There are hundreds of identified *E. coli* strains that result in a number of diseases ranging from mild, self-limiting to aggressive type and sepsis (*Hassan, Ojo & Abdulrahman, 2021*). *E. coli* remains the

leading bacterial pathogen for many common infections, such as urinary tract infection (UTI), diarrhea, and neonatal sepsis with multidrug resistance due to extended-spectrum beta-lactamases (ESBL) production (*Zhu et al., 2019*).

Resistance to different classes of antimicrobial agents among ESBL producers at the same time was primarily detected in wound infections as well as other infections. Aquatic environments are considered a significant source for disseminating multidrug-resistant (MDR) bacteria, basically *E. coli*, with transferring antibiotic-resistant genes through plasmid mainly in the hospital environment, which puts an extra burden on the hospital (*Furlan, Savazzi & Stehling, 2020*). *E. coli* has a remarkable and complex genomic plasticity for taking up and accumulating genetic elements; thus, multiresistant high-risk clones can evolve (*Kocsis, Gulyás & Szabó, 2022*). *E. coli* clone sequence type (ST) 131 is the predominant pathogen of extra intestinal source that harbors ESBL genes, specifically the CTX-M family. This clone belongs to the virulent phylogenetic group B2, with serotype O25: H4, especially O25b (*Al-Guranie & Al-Mayahie, 2020*).

Among the large family of ESBL, cefotaximases ($bla_{\text{CTX-M}}$) are becoming a significant group that is globally disseminated. Both $bla_{\text{CTX-M-15}}$ and $bla_{\text{CTX-M-14}}$ are the predominant ESBL genes that restrict treatment options and demand the need for the carbapenem group as the antibiotic of choice for treatment; so far, increasing the rate of spread and dissemination of carbapenemase-producing Enterobacteriaceae (*Emeraud et al., 2019*). With an increasing prevalence of ESBL enzyme ($bla_{\text{CTX-M-15}}$), there is a simultaneous resistance against fluoroquinolone, especially in extraintestinal pathogenic *E. coli* (*Szijártó et al., 2015*). This causes most hospitals to acquire nosocomial infection with sequence type 131 (ST131)-O25B: H4 that might have multidrug-resistant properties and be able to spread globally (*Morales-Barroso et al., 2017*).

The clonal dissemination of $bla_{\text{CTX-M-15}}$ of ESBL, which causes different infections, is primarily harboured by homogenous plasmids originating from contaminated water and commercial food, and in some countries, the source of dissemination is travelling (*Kurittu et al., 2021*).

The pandemic of SARS-CoV-2 infection at the beginning of 2020 has heavily hit most countries in the world, and one of the significant challenges imposed by this infection has been the large number of patients hospitalized due to the severity of the disease. Bacterial infection, especially *E. coli*, has been reported in hospitalized patients with COVID-19 (*Grasselli et al., 2021*), but there is limited experience with these infections in our locality (*Bardi et al., 2021*). Thus, this study aimed to determine the prevalence of O25B-ST131clone with molecular detection of $bla_{\text{CTX-M-15}}$ ESBL among *E. coli* from different samples of hospitalized patients with COVID-19.

## MATERIALS AND METHODS

### Study design and setting

This cross-sectional study was conducted on 528 patients who were admitted to Anwar Shexa Medical City, Sulaimaniyah, Iraq, and hospitalized from April to September 2021 because of COVID-19 with various symptoms (Supplementary Files).

### Inclusion criteria

Hospitalized infected patients with COVID-19 regardless of age and gender.

### Exclusion criteria

Out-patients were excluded from the study.

### Sample collection and analysis

Patient samples ($n = 528$) were collected from different infection sites, including urine, catheter tip, sputum, blood, endotracheal tube aspiration (ETA), pleural fluid (PF), peritoneal fluid, and throat swabs. Then, samples were processed through standard bacteriological techniques using blood agar, MacConkey agar and eosin methylene blue (*Dadi et al., 2018*). A sample was considered culture-positive for *E. coli* isolate when the organism was detected at a concentration of $>10^5$ colony-forming units (CFU) (*Shaik et al., 2017*).

### Antibiotic susceptibility test

It was performed through the BD Phoenix™ M50 system using panels for Gram-negative identification and the antibiotic susceptibility profile according to the manufacturer's recommendation (*Hong et al., 2019*).

### ESBL production test
#### Double disc synergy test (DDST)

All isolated *E. coli* resistant to second and third-generation cephalosporin were subjected to ESBL production using DDST according to CLSI 2021 (*Wayne, 2020*). Briefly, antibiotic discs of ceftazidime (CAZ, 30 μg), cefotaxime (CTX, 30 μg), ceftriaxone (CTR, 30 μg), cefixime (CFM, five μg), and cefepime (CPM, 30 μg) were placed at a distance of 20 mm from center on lawn cultures on Muller Hinton agar plates, and a combined amoxicillin (20 μg)/clavulanic acid (10 μg) disc (AMC, 30 μg) was placed at the center of plate and were incubated overnight at 37 °C. Any enhancement in the zone of inhibition around each cephalosporin disc towards the AMC disc was considered to be ESBL positive (Fig. 1).

#### Combined disc test (CDT)

A disc of clavulanic acid (CEC, 30 μg) and cefotaxime disc (CTX, 30 μg) were used together. The CTX disc was placed at a distance of 20 mm apart from the CEC disc on a lawn culture of the resistant isolate on the Muller Hinton agar plate and was incubated overnight at 37 °C. The *E. coli* isolate was considered ESBL positive when the diameter zone around the combined disc is $> 5.0$ mm (*Salihu, Yarima & Atta, 2020*) (Fig. 2).

### Molecular detection of O25B-ST131 clone and *bla*$_{CTX-M-15}$ genes

According to manufacturer instructions, DNA extraction was performed for all isolated *E. coli* using a MagPurix reagent kit (Zinexts Life Science Corporation, New Taipei City, Taiwan). Then, extracted DNA samples were screened through Real-Time PCR to detect the O25b-ST131 clone and *bla*$_{CTX-M-15}$ genes of *E. coli* with different cycling conditions (Figs. 3 and 4) (*Kurittu et al., 2021*). Briefly, PCR reaction was prepared with a master mix

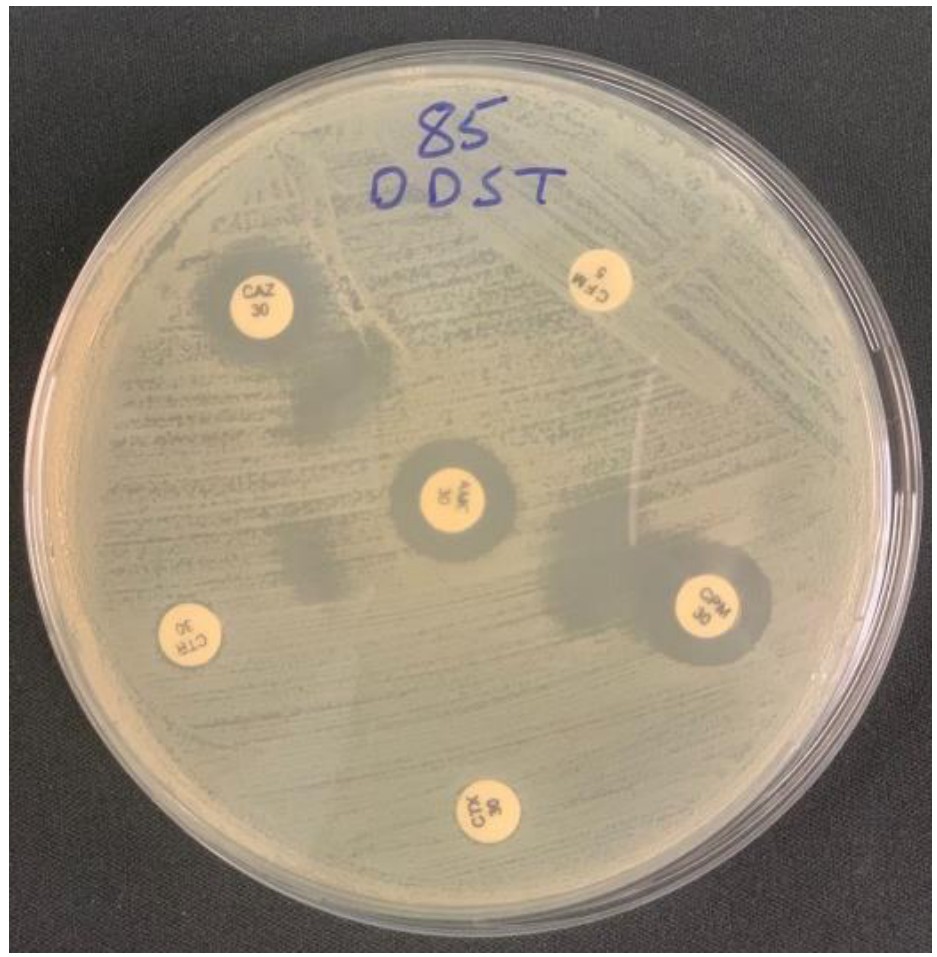

**Figure 1** The double disc synergy test shows an extension zone of inhibition from amoxicillin (20 μg)/clavulanic acid (10 μg) disc (AMC, 30 μg) towards discs of second and third generation cephalosporin, including ceftazidime (CAZ, 30 μg), cefotaxime (CTX, 30 μg), ceftriaxone (CTR, 30 μg), cefixime (CFM, 5 μg), and cefepime (CPM, 30 μg).

(SYBR Green, Thermo Fisher Scientific, Milton Park, UK) containing 1.0 μL from each F and R primer (*Demirci, Ünlü & Tosun, 2019*) (Table 1), 5.0 μL DNA templates, 0.3 μL DNA polymerase, and then completed to 25 μL with ddH2O. Then, the PCR program started with initial denaturation at 94 °C for 10 min, 30 cycles of denaturation at 94 °C for 40 s, annealing at 55 °C for 40 s, extension at 72 °C for 1.0 min, and a final extension at 72 °C for 7.0 min. *E. coli* ATCC 25922 was used as a negative control.

## Gene sequencing

The obtained PCR products of two *E. coli* clones that were positive were sent to the Macrogen Genome Center Company, Republic of South Korea, for $bla_{CTX-M-15}$ gene sequencing using Sanger sequencing to confirm the nucleotide identity at both ends of the amplicon.

**Figure 2** Combined disc test shows a zone of inhibition larger than 5.0 mm around clavulanic acid disc (CEC, 30 $\mu$g) compared to small inhibition zone around cefotaxime disc (CTX, 30 $\mu$g).

**Table 1 Primer sequence of O25b-ST131 clone and $bla_{CTX-M-15}$ genes.**

| Gene | Primer sequence | Reference |
|------|-----------------|-----------|
| O25B-ST131 clone | ST131TF (5-GGT GCT CCA GCA GGT G-3) | *Demirci, Ünlü & Tosun (2019)* |
| | ST131TR (5-TGG GCG AAT GTC TGC-3) | |
| $bla_{CTX-M-15}$ | MC-3-15F (5-TGG GGG ATA AAA CCG GCA G-3) | *Demirci, Ünlü & Tosun (2019)* |
| | MC-3-15R (5-GCG ATA TCG TTGGTG GTG C-3) | |

## Statistical analysis

Data analysis was performed using Statistical Package for Social Sciences (SPSS, version 27.0; SPSS Inc., Chicago, IL, USA). The data were expressed as numbers and percentages. The chi-square test was used for categorical variables. The overall percentage of agreement was calculated, and the Cohen's kappa ($\kappa$) statistic value was estimated. A *p*-value of < 0.05 was considered a significant difference.

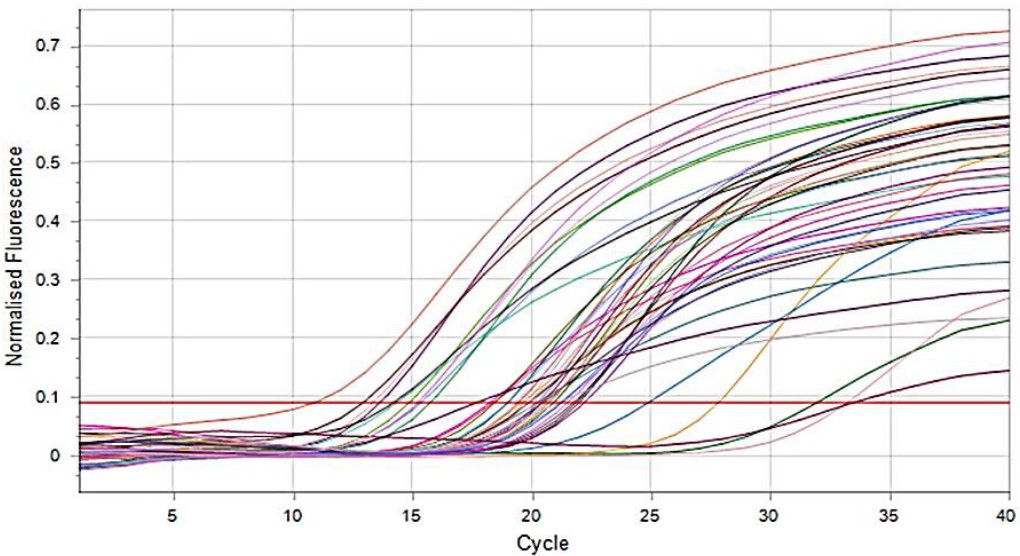

**Figure 3 SYBR green real time PCR amplification result of ST131 clone *E. coli* (fluorescent *versus* cycle number).** The figure outlines the relationship between the amount of DNA (ST131 clone *E. coli*) measured and the amplification cycle. All tested samples show increasing copy numbers.

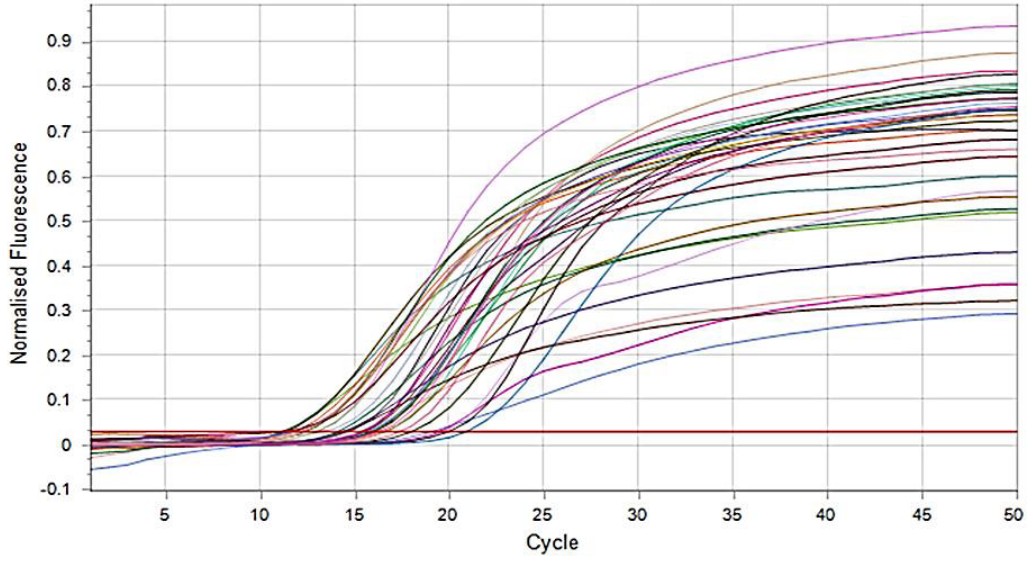

**Figure 4 SYBR green real time PCR amplification result of *bla*$_{CTX-M-15}$ (fluorescent *versus* cycle number).** The figure outlines the relationship between the amount of DNA (*bla*$_{CTX-M-15}$) measured and the amplification cycle. All tested samples show increasing copy numbers.

## RESULTS

Among 528 processed COVID-19 patients' samples, only 234 samples were positive for *E. coli* isolates using various culturing media. Antibiotic susceptibility test revealed that all

isolated *E. coli* ($n = 234$) were resistant to all used antibiotics ($n = 13$), but with various values, as most isolates (79.9%) were resistant to CTR, followed by CAZ (70.9%) and then CFE (70.5%). However, 97.9% of isolates were sensitive to CL, 90.2% to MEM, 89.3% to AK, and 86.7% to IMP (Table 2). Consequently, *E. coli* isolates that were resistant to at least two of the three cephalosporins (cefotaxime, ceftriaxone and ceftazidime) were tested for ESBL production. Thus, the phenotypic test showed that 63.2% ($n = 148$) of *E. coli* isolates were positive for ESBL, while the other 36.8% ($n = 86$) were negative using screening tests (DDST, CD1 and CD2). Among the positive ESBL isolates ($n = 148$), 58.1% ($n = 86$) were confirmed by DDST, 83.8% ($n = 124$) and 60.1% ($n = 89$) by CDT1 and CDT2, respectively. On the other hand, among the negative ESBL isolates ($n = 86$), 62.8% ($n = 54$) were confirmed by DDST, 60.5% ($n = 52$) and 65.1% ($n = 56$) were confirmed by CDT1 and CDT2, respectively. The difference between each of the phenotypic tests was significant ($p = 0.002$, $p < 0.001$ and $p < 0.001$ for DDST, CDT1 and CDT2, respectively) (Table 3). Moreover, we calculated the estimated overall percentage agreement from Table 3 data as follows: Overall percentage agreement $(a + d)/(a + b + c + d) \times 100\%$. For example, for DDST $= (54 + 86)/(54 + 62 + 32 + 86) \times 100\% = (140/234) \times 100\% = 59.8\%$, with kappa value of 0.20. Consequently, the percentage agreement for CDT1 and CDT2 were 72.5% and 65.8%, with kappa values of 0.24 and 0.23, respectively. Based on the genotypic test, results revealed that 68.4% ($n = 160$) of *E. coli* isolates were positive for the *bla*$_{CTX-M-15}$ gene, and the other 31.6% ($n = 74$) were negative. Among the positive *bla*$_{CTX-M-15}$ gene, 25% ($n = 40$) of *E. coli* isolates were negative for the O25B-ST131 clone, and 75% ($n = 120$) were positive whereas, out of 74 negative *bla*$_{CTX-M-15}$ genes, 25.7% ($n = 19$) of *E. coli* isolates were negative for O25B-ST131 clone, and 74.3% ($n = 55$) were positive without significant difference ($p = 0.912$) (Table 4). In comparison of the CDT1 to genotypic test, among the 160 positive *bla*$_{CTX-M-15}$ genes, 80.6% ($n = 129$) of *E. coli* isolates were confirmed by CDT1, while the other 19.4% ($n = 31$) were negative. On the other hand, among the 74 negative *bla*$_{CTX-M-15}$ genes, 30.2% ($n = 29$) of *E. coli* isolates were positive for CDT1 and the other 60.8% ($n = 45$) were negative. Thus, the association between genotypic and phenotypic tests in regard to *bla*$_{CTX-M-15}$ gene detection is significant ($p < 0.001$) (Table 5). Finally, the results of the gene sequencing of the *E. coli* O25B-ST131 clone positive was compatible with the desired gene and was accepted by NCBI Gene Bank with an accession number of OP713886.

## DISCUSSION

*Escherichia coli* ST131 is a crucial multidrug-resistant clone responsible for more than half of ESBL-producing *E. coli* isolates. In this study, we aimed to investigate the presence of O25b-ST131 clone and *bla*$_{CTX-M-15}$ genes in the *E. coli* strains isolated from hospitalized COVID-19 patients using real-time PCR. Thus, this work is conducted in the research area for the first time in Iraq and showed the prevalence of genes in *E. coli* from our locality.

Antimicrobial susceptibilities of the isolated *E. coli* ($n = 234$) showed various degrees of resistance to currently used common antibiotics, such as CTR (79.9%), CAZ (70.9%), and CFE (70.5%). However, 97.9% of isolates were sensitive to CL, 90.2% to MEM, 89.3%

**Table 2  Antibiotic susceptibility profile of isolated *E. coli*.**

| Antibiotic disc | *E. coli* susceptibility ($n = 234$) Number (%) | | |
|---|---|---|---|
| | Intermediate | Resistant | Sensitive |
| AMC | 35 (15.0) | 100 (42.7) | 99 (42.3) |
| PZT | 21 (9.0) | 36 (15.4) | 177 (75.6) |
| CTR | 2.0 (0.9) | 187 (79.9) | 45 (19.2) |
| CAZ | 11 (4.7) | 166 (70.9) | 57 (24.4) |
| CFE | 17 (7.3) | 165 (70.5) | 52 (22.2) |
| IMP | 5.0 (2.1) | 26 (11.2) | 203 (86.7) |
| MEM | 4.0 (1.7) | 19 (8.1) | 211 (90.2) |
| CL | 3.0 (1.3) | 2.0 (0.9) | 229 (97.9) |
| CIP | 4.0 (1.7) | 147 (62.8) | 83 (35.5) |
| LEV | 3.0 (1.3) | 147 (62.8) | 84 (35.9) |
| STX | 1.0 (0.4) | 131 (56.0) | 102 (43.6) |
| AK | 3.0 (1.3) | 22 (9.4) | 209 (89.3) |
| CN | 2.0 (0.9) | 53 (22.6) | 179 (76.5) |

Notes.

AK, Amikacin; AMC, Amoxicillin and Clavulanic Acid; CAZ, Ceftazidime; CFE, Cefipime; CIP, Ciprofloxacin; CL, Colistin; CN, Gentamycin; CTR, Ceftriaxone; IMP, Imipenem; LEV, Levofloxacin; MEM, Meropenem; PZT, Piperacillin-Tazobactam; STX, Trimethoprim-sulfamethoxazol.

**Table 3  Association of both phenotypic tests and screening test for ESBL.**

| Phenotypic test | | Screening test for ESBL Number (%) | | *p*-value |
|---|---|---|---|---|
| | | Negative 86 (36.8) | Positive 148 (63.2) | |
| DDST | **Negative** 116 (49.6) | 54 (62.8) | 62 (41.9) | 0.002[*] |
| | **Positive** 118 (50.4) | 32 (37.2) | 86 (58.1) | |
| CDT 1 | **Negative** 76 (32.5) | 52 (60.5) | 24 (16.2) | <0.001[**] |
| | **Positive** 158 (67.5) | 34 (39.5) | 124 (83.8) | |
| CDT 2 | **Negative** 115 (49.1) | 56 (65.1) | 59 (39.9) | <0.001[**] |
| | **Positive** 119 (50.9) | 30 (34.9) | 89 (60.1) | |

Notes.

DDST, Double disc synergy test; CDT, Combined disc test; EBSL, Extended-spectrum beta-lactamases.

[*] Significant difference.

[**] Highly significant difference using chi-square test.

**Table 4  Number and percentage of O25B-ST131 clone of *E. coli* and *bla*$_{\text{CTX-M-15}}$ gene.**

| *E. coli* O25B-ST131 clone | *bla*$_{\text{CTX-M-15}}$ gene Negative 74 (31.6%) | *bla*$_{\text{CTX-M-15}}$ gene Positive 160 (68.4%) | Total | *p*-value (chi-square test) |
|---|---|---|---|---|
| | | Number (%) | | |
| **Negative** | 19 (25.7) | 40 (25.0) | 59 (25.2) | 0.912 |
| **Positive** | 55 (74.3) | 120 (75.0) | 175 (74.8) | |

**Table 5  Association between genotypic and phenotypic tests in regard to $bla_{CTX-M-15}$ gene.**

| CDT1 | Genotypic test Number (%) | | p-value |
| --- | --- | --- | --- |
| | $bla_{CTX-M-15}$ gene Negative 74 (31.6) | $bla_{CTX-M-15}$ gene Positive 160 (68.4) | |
| Negative 76 (32.5) | 45 (60.8) | 31 (19.4) | <0.001[**] |
| Positive 158 (67.5) | 29 (39.2) | 129 (80.6) | |

Notes.

CDT, Combined disc test.

[**]Highly significant difference using Chi-Square test.

to AK, and 86.7% to IMP. In this regard, another study realized that carbapenems were the best therapeutic choice for all *E. coli* isolates, followed by nitrofurantoin, amikacin and fosfomycin (*Demirci, Ünlü & Tosun, 2019*). *Namaei et al. (2017)* reported *E. coli* sensitivity to AK, AMC, CFE, CTX, CN, STX, CIP, IMP, and MEM as 97.2%, 88.4%, 79.4%, 33.4%, 67.7%, 29.9%, 23.2%, 0% and 0%, respectively. These disparities among studies might be related to the sample size, geographical location, samples collected from patients with various diseases, the quality of the antibiotic disc, and the manufacturer companies of the used disc. Generally, it appeared likely that during the COVID-19 pandemic, antimicrobial resistance rates would be higher because of the overutilization of antimicrobials for patients with COVID-19, with or without secondary infection, and because of the increased rate of hospitalization associated with COVID-19 (*Lai et al., 2021*).

Extended spectrum beta-lactamase (ESBL) producers are still the leading cause of infections among populations and directly affecting the public health measures. They confer resistance to most beta-lactam antibiotics including the third generation cephalosporins and monobactam antibiotics sparing the cephamycins. Infections with these ESBL-producing organisms have been associated with poor outcomes. Thus, phenotypic screening and confirmatory tests to identify the ESBL producer are crucial to the clinical management. In this study, the phenotypic test showed that 63.2% of *E. coli* isolates were positive for ESBL; among them, 58.1% were confirmed by DDST ($p = 0.002$), 83.8% by CDT1 ($p < 0.001$) and 60.1% by CDT2 ($p < 0.001$). These results are agreed with that of *Demirci, Ünlü & Tosun (2019)* who found that 66.67% of *E. coli* isolates were producing ESBL. On the other hand, *Lemenand et al. (2021)* in France stated a decreasing proportion of ESBL among *E. coli* infections during the COVID-19 pandemic, while *Hasan et al. (2023)* in Canada studied 2.3 million positive urine cultures and found 48.9% *E. coli* isolates of which 5.8% were produced ESBLs that was higher in the pandemic period than in the pre-pandemic period. These variations among studies might be associated with the sample size, rate of COVID-19 infection, number of hospitalized patients, severity of the disease, and type of the test used for detection, as in this study, we found that the CDT1 is a better phenotypic screening test than DDST and CDT2 for detection of ESBL producer *E. coli*.

According to the genotypic tests using molecular assay, our results revealed that 68.4% of ESBL-producing *E. coli* isolates were positive for the $bla_{CTX-M-15}$ gene. Among them, 75% were positive for O25B-ST131 clone. However, among *E. coli* isolates with negative

$bla_{CTX-M-15}$ gene, 74.3% were positive for O25B-ST131 clone. *Al-Guranie & Al-Mayahie (2020)* in Wasit City, Iraq, found 92.1% (35/38) of *E. coli* isolates were positive for ST131, of which 34 was the O25b-ST131 strain and one was the O16-ST131 strain. Another study in Turkey in 2019 found the positive $bla_{CTX-M-15}$ gene and O25b-ST131 clone to be 35.3% and 31.37%, respectively, among the phenotypically *E. coli* ESBL positive strains, which is quietly less percentage than that reported in the current study (*Demirci, Ünlü & Tosun, 2019*). In contrast, *Namaei et al. (2017)* in Iran reported 78.5% for the O25b-ST131 clone and 95.5% for the $bla_{CTX-M-15}$ gene in ESBL-positive *E. coli* strains. A study by *Marialouis & Santhanam (2016)* in India reported that 47% of *E. coli* strains were ESBL producers and 41% had O25b-ST131 clones. These variations in the results might be related to using different molecular techniques with varying protocols in these studies.

The *E. coli* O25B-ST131 clone is considered an important public health problem due to its epidemic potential, virulence and multidrug resistance ability, which were dramatically higher than non-O25B-ST131 clones (*Dautzenberg et al., 2016*). The presence of an O25B-ST131 clone without ESBL in commensal and pathogenic conditions could represent a potential threat to further emergence of resistance; however, the O25B-ST131 clone is reported to be strongly associated with ESBLs such as the producing of $bla_{CTX-M-15}$ (*Mesquita et al., 2021*). Hence, our results confirmed that O25B-ST131 clone production is not potentially related to $bla_{CTX-M-15}$ gene mutation, as strains with negative $bla_{CTX-M-15}$ gene also had O25B-ST131 clones. This leaves limited therapeutic options for the treatment of infections caused by this clone and increases the interest in the monitorization of this infectious agent (*Namaei et al., 2017*). Accordingly, *the E. coli* O25B-ST131 clone positive was compatible with the desired gene and given an accession number of OP713886 by NCBI Gene Bank.

## CONCLUSIONS

All isolated *E. coli* were resistant to the current commonly used antibiotics. CDT1 is better than DDST and CDT2 tests as a phenotypic screening test for ESBL detection in *E. coli* isolates. Most isolates were ESBL producers, held $bla_{CTX-M-15}$ gene and were positive for the O25B-ST131 clone. Thus, the detection of $bla_{CTX-M-15}$ gene-producing *E. coli* clone O25B-ST131 among hospitalized COVID-19 patients in Sulaimaniyah, Iraq, needs more action from the hospital infection control teams. This study has several limitations, including the sample collection from one hospital rather than the whole city hospitals, as there are a limited number of hospitals specified for admitting COVID-19 patients and more study needed to illustrate the details of clones through multi-sequence typing.

## ACKNOWLEDGEMENTS

I thank the bacteriology staff members of High Quality and Molecular Laboratories at Anwar Shexa Medical City, Sulaimaniyah, Iraq for their help with the culture media preparation.

### Funding

The author received no funding for this work.

### Competing Interests

The author declares that they have no competing interests.

### Author Contributions

- Khanda Abdulateef Anwar conceived and designed the experiments, performed the experiments, analyzed the data, prepared figures and/or tables, authored or reviewed drafts of the article, and approved the final draft.

### Human Ethics

The following information was supplied relating to ethical approvals (*i.e.*, approving body and any reference numbers):

The study protocol was approved by the Ethical Committee of the College of Medicine, University of Sulaimani, Sulaimaniyah, Iraq (No. 242/10/19/12/2021-UoS). The Anwar Shexa Medical City Intuitional Review Board (IRB), Sulaimaniyah, Iraq also approved method section of this study (No. 03 on March 20, 2021).

### DNA Deposition

The following information was supplied regarding the deposition of DNA sequences:

Data is available at GEO: OP713886.1.

### Data Availability

Raw data are available in the Supplemental Files.

### Supplemental Information

Supplemental information for this article can be found online at http://dx.doi.org/10.7717/peerj.19011#supplemental-information.

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
