# Peer review of "Detection of O25B-ST131 clone and blaCTX-M-15 gene in Escherichia coli isolated from patients with COVID-19"

_PeerJ, doi:10.7717/peerj.19011_

## Round 0.1 · original submission · Major Revisions

Dear Authors,

Thank you for submitting your manuscript, titled "Detection of O25B-ST131 clone and blaCTX-M-15 gene in Escherichia coli isolated from patients with COVID-19," to PeerJ. The reviewers have provided detailed and constructive feedback to improve the manuscript. Based on their reports, we believe the study has the potential to make a valuable contribution but requires significant revisions before it can be considered further. Please send us point-by-point responses to move further.

Best wishes,
Dr. Nagendran Tharmalingam
Handling Editor.

Reviewer 1 ·

Basic reporting

Some language use was unclear or misleading, inconsistency in terms (Kappa, DDST, CST) throughout the paper. Raw data that was shared is not representative of results from the study (no E. coli species listed). Many initial tables/figures should be shared as supplemental information.

Experimental design

Research aim is a little muddled, apart from estimating prevalence, it seems a lot of the results are focused on characterizing the three ESBL detection methods- which is an important finding and not rigorously discussed. Methods are lacking key details, and sample inclusion criteria for testing is not clear. The authors will need to make clear in the methods the flowpath from the number of isolates of E. coli > number resistant and fit the inclusion criteria for ESBL testing > the number that were sent for confirmation with DDST, CDT1 and CDT2 > number of isolates sent for gene sequencing.

Validity of the findings

The raw data shared is not representative. Results and conclusions are stated but not fully explored in the discussion with context to the population and how this will impact or support public health measures.

·

Basic reporting

The authors have provided the article with clear language and the Language delivery is properly understandable. The references are sufficient, although reference number 8, 9, 13, and 25 is older than 10 years from its publication. A more recent reference is needed to replace reference number 8 as the reference is needed for its prevalence.
The article is well structured and well described. The association between O25B-ST131 clone and blaCTX-M-15 gene is not statistically significant, but can imply the dominance of the sequence type in the prevalence of ESBL.
Figures and tables are relevant. The raw data is also provided.
The article is self-explanatory and the results are relevant to the discipline.

Experimental design

This original primary research fits the aims and scope of PeerJ. The research question that arises from reading the title and abstract is well answered throughout the text. the question is also meaningful. the methods that was used to detect O25B-ST131 clone and bla CTX-M-15 gene is feasible and mostly applicable.
The detailed description of the methods can be reproduced by another research..
Ethical considerations does not imply any problem.

The raw data has been provided, however the supplemental files need to be called out in the text according to its context. A more descriptive identifiers is needed in the main text to be useful to future readers.

Validity of the findings

The replication of this study is encouraged. The identification of O25B-ST131 clone and bla CTX-M-15 gene using PCR and SYBR is replicable and needed to be performed in many health care facilities throughout the world.
All underlying data is shared as supplementary material and statistically sound. The authors presents enough data and the data is enough to formulate the conclusions and well explained in the discussion.
Conclusions are well stated and answers the original research questions.

Additional comments

Pay attention to mistyping such as "Kappa", etc. Formulas should be written using formula feature (Line 140-141).

·

Basic reporting

1) Manuscript is written in a very clear language, easy to understand and all experiments linked to each other.
2) References are relevant; however more latest citations may improve the impact of study to show that this study provides new data set OR compare and contrast with other countries data would be more impactful.
3) All data is shared, tables and figures are constructed in good way, easy to understand.

Experimental design

Experimental design is sufficient to support the findings of the study. All experiments are relevant and justify the obtained results. Author may mention in the abstract that this is first study of its kind from study area and no such study has been conducted in the study area. This will enhance the impact and show the situation of E.Coli in the study area specifically.

Validity of the findings

Data of this study is good, however, author did not emphasize the novelty of the study that it is first time conducted in the research area and showed the prevalence of genes in E.coli from research area.

Additional comments

Please add few sentences to show novelty of the study in the abstract conclusion part.

---

## Round 0.2 · accepted · Accept

Dear Authors,

We are pleased to inform you that your work has been accepted for publication in PeerJ. The production team will contact you soon regarding any typesetting questions. Thank you for submitting your work, and we look forward to seeing your next submission.

Best wishes,
Dr. Nagendran Tharmalingam
Handling Editor.

·

Basic reporting

The authors have provided the article with clear language and the Language delivery is properly understandable. The article is well structured and well described. The association between O25B-ST131 clone and blaCTX-M-15 gene is not statistically significant, but can imply the dominance of the sequence type in the prevalence of ESBL. Figures and tables are relevant. The raw data is also provided. The article is self-explanatory and the results are relevant to the discipline.

Experimental design

This original primary research fits the aims and scope of PeerJ. The research question that arises from reading the title and abstract is well answered throughout the text. The question is also meaningful. The methods that was used to detect O25B-ST131 clone and bla CTX-M-15 gene is feasible and mostly applicable. The detailed description of the methods can be reproduced by another research. Ethical considerations doesnot imply any problem.

Validity of the findings

Validity of the findings The replication of this study is encouraged. The identification of O25B-ST131 clone and blaCTX-M-15 gene using PCR and SYBR is replicable and needed to be performed in many health care facilities throughout the world. All underlying data is shared as supplementary material and statistically sound. The authors present enough data and the data is enough to formulate the conclusions and well explained in the discussion. Conclusions are well stated and answers the original research questions.